# Anastomotic Urethroplasty with Double Layer Continuous Running Suture Re-Anastomosis Versus Interrupted Suture Re-Anastomosis for Infective Bulbar Urethral Strictures: A Prospective Randomised Trial

**DOI:** 10.3390/jcm11154252

**Published:** 2022-07-22

**Authors:** Frederik M. Claassen, Francisco E. Martins, Shingai B. A. Mutambirwa, Linda Potgieter, Lezelle Botes, Harry F. Kotze, Francis E. Smit

**Affiliations:** 1Department of Urology, Faculty of Health Sciences, University of the Free State, Bloemfontein 9301, South Africa; claassen@ufs.ac.za; 2School of Medicine, University of Lisbon, Santa Maria Hospital, 1649-004 Lisboa, Portugal; 3Department of Urology, Sefako Makgatho Health Sciences University, Dr George Mukhari Academic Hospital, Ga-Rankuwa 0208, South Africa; sbmutambirwa@hotmail.com; 4Department of Cardiothoracic Surgery, Faculty of Health Sciences, University of the Free State, Bloemfontein 9301, South Africa; potgieter.linda@gmail.com (L.P.); hf@ufs.ac.za (H.F.K.); smitfe@ifs.ac.za (F.E.S.); 5Department of Health Sciences, Faculty of Health and Environmental Sciences, Central University of Technology, Bloemfontein 9301, South Africa; lbotes@cut.ac.za

**Keywords:** urethral stricture, anastomotic urethroplasty, bulbar urethral stricture

## Abstract

Introduction: The objective of this study was to compare a double-layer running suture re-anastomosis urethral stricture repair with early catheter removal to the conventional interrupted suture re-anastomosis after excision of a bulbar urethral stricture. Methods: A consecutive series of patients with bulbar urethral stricture were enrolled in the study. The patients were randomized into two groups according to an odd/even serial number distribution. Patients’ medical records were analyzed for demographics, stricture characteristics, and lower urinary tract obstructive symptoms. The outcomes were based on the presence/absence of obstructive voiding symptoms, and retrograde urethrography (RGU) performed on the first post-operative day in Group 1 and in both groups (Groups 1 and 2) at six weeks after surgery. Flexible urethroscopy was only performed on specific cases where RGU was unclear both pre- and post-operatively or when clinical recurrence was suspected. The minimum follow-up (FU) was 18 months. Success was defined as no need for subsequent dilatation, direct vision internal urethrotomy (DVIU), or urethroplasty. Results: A total of thirty-six patients with a mean age of 45 years (range 20 to 69 years) with bulbar urethral stricture were included in this study. Group 1 and Group 2 included 19 and 17 patients, respectively. Two patients were lost during randomization and subsequently to FU. The average stricture lengths were comparable between the two groups according to the retrograde urethrogram: 1.20 cm (range 0.6 to 2) in Group 1 and 1.27 cm (range 0.5 to 2.4) in Group 2, respectively (*p* = 0.631). The success rate for Group 1 was 90% after a mean follow-up of thirty-six months (range 20 to 40), which was clinically significant compared to the 71% in Group 2 after a mean FU of thirty-three months (range 19 to 40; *p* = 0.0218; 95% CI: 0.462–41.5766). Conclusions: Anastomotic urethroplasty (AR) performed with a double layer re-anastomosis had a cure rate comparable to the conventional anastomosis with interrupted sutures after a follow-up of eighteen months and longer. The urethral catheter can be safely removed within twenty-four hours after the excision of stricture and double-layer re-anastomosis.

## 1. Introduction

Anastomotic urethroplasty (AU) for bulbar urethral strictures has set a gold standard for strictures less than 2 cm [1]. AU yields good results for iatrogenic or trauma-related bulbar urethral strictures shorter than 2 cm, with a reported success rate of approximately 95% [2]. Elthahawy et al. (2007) report a success rate of 99% in a series of 260 patients who underwent AU for traumatic and instrument-related strictures [3]. One of the advantages of AU is that patients can be discharged early post-operatively, even as a same-day procedure [4]. This surgical technique involves using a watertight double-layer running suture re-anastomosis and the removal of the urethral catheter upon discharge on day one post-operatively. The early removal of the urethral catheter may decrease catheter-related problems, such as urinary tract infections and possible pressure necrosis of the urethral epithelium, which increase the risk of re-stricture [5]. The aim of this study was to compare the outcomes of the double layer running suture anastomosis with early catheter removal to the conventional interrupted suture technique with fourteen-day catheterization in a prospective randomized study.

## 2. Methodology

A consecutive series of patients with bulbar urethral stricture were enrolled in the study. The patients were randomized into two groups according to an odd/even serial number distribution. Patients’ medical records were analyzed for demographics, stricture characteristics, and lower urinary tract obstructive symptoms. The preoperative evaluation included clinical history, physical examination, urine analysis, and RGU to determine the length and location of the urethral stricture. Flexible urethroscopy was only performed in specific cases where urinary symptoms and RUG were unclear. The RGU was performed using a clamp method with drip infusion as described in the literature [5]. The inclusion criteria were stricture lengths ≤ 2.5 cm, located in the bulbar urethra, and patients with ≤1 previous DVIU. All patients with >1 DVIU and previous urethroplasty failures were excluded.

The patients were allocated to 2 groups according to an odd/even serial distribution upon their arrival at the clinic (G1 = odd numbers; G2 = even numbers). A prophylactic broad-spectrum intravenous antibiotic (e.g., 1 g ceftriaxone, was administered preoperatively.

All of the surgical procedures were performed under general anaesthesia and by the same surgeon (FMC). The patients were placed in the lithotomy position, and the urethra approached through a perineal incision over the length of the urethral stricture splitting the corpus spongiosum muscle. The urethra was probed with a 16Fr catheter to determine the location of the stricture and then mobilized from the corpora cavernosum and transected in the stricture site. The urethral stricture was excised, and the healthy urethral ends were sutured with 4/0 Vicryl (polyglactin) running sutures in two layers in Group 1 (Figure 1 and Figure 2). The double-layer continuous running suture 4/0 Vicryl re-anastomosis was performed by first suturing the urethral mucosa, starting from a 12 o’clock position with two sutures closed in a running fashion clockwise and anti-clockwise to the 6 o’clock position. The spongiosial plane was also sutured in a similar fashion as the mucosal one with 4/0 Vicryl reabsorbable material (Figure 1 and Figure 2). The anastomosis after stricture excision in patients of Group 2 was performed with five interrupted sutures incorporating the epithelium and spongiosum with 3/0 Vicryl sutures (Figure 3).

After the completion of the anastomosis, the bulbospongiosum muscle was re-approximated over the urethra with a 4/0 Vicryl suture and the skin was then closed with a 3/0 Vicryl running suture in both groups.

The urethral catheters of patients in Group 1 were removed on day one and after fourteen days in Group 2. After urethroplasty, RGU was performed at the time of catheter removal, and any occurrence of contrast extravasation was documented. All of the patients were re-evaluated with RUG at six weeks post-operatively. Patients in Group 1 were discharged on day one with saline sitz baths and oral analgesics after urethral catheter removal. The urethral catheters in Group 2 were removed 14 days following surgery. All of the patients were then followed up at a 3-month interval for an 18-month period with a focus on the detailed history of urethral stricture recurrence. Success was defined as no need for subsequent dilatation or any adjunctive endoscopic instrumentation. 

### Statistical Analysis

The Student’s test was used to compare age and length. The categorical data were analyzed with the chi-squared test and Fisher’s exact test. Differences were regarded as significant with *p* ≤ 0.05. All of the calculations were conducted with SPSS^®^ release 15. (SPSS Inc, Chicago, IL, USA).

## 3. Results

A total of thirty-six patients with a mean age of 45.5 years (range 20 to 69) with bulbar urethral stricture were included in this study. All of the patients had a history of purulent urethral discharge. Five patients had received treatment for a gonococcal infection elsewhere prior to referral to our institution, and thirty-one patients were diagnosed and treated for confirmed gonococcal infection at our institution.

The mean stricture length, as assessed by RUG, was 1.20 cm (range 0.6 to 2) in Group 1, and 1.27 cm (range 0.5 to 2.4) in Group 2, respectively (*p* = 0.631) (Table 1). The overall success rate after an 18-month minimum FU period for Group 1 was 90% compared to the 71% in Group 2 (*p* = 0.2185; CI 95%; 0.4628 to 41.5756).

The RGU performed at the time of catheter removal of patients in Group 1 on day one demonstrated minimal extravasation but no voiding difficulty in six patients (32%). These patients were discharged from the hospital without reinserting a Foley catheter. The extravasation cleared up spontaneously when the patients had a follow-up RGU six weeks after catheter removal (Figure 4 and Figure 5). The patients with minimal extravasation did not develop any clinical problems. The patients in Group 2 had their urethral catheters removed two weeks post-surgery, and an RGU was performed six weeks after catheter removal. Stricture recurrence occurred in Group 1 at 3.6 and 6.2 months. The seven failures in Group 2 occurred at an average of 5.6 months (range 1.1 to 10.1) (*p* = 0.804).

## 4. Discussion

This study demonstrated that a short hospital stay with catheter removal on the first post-operative day is feasible after performing a double running suture anastomotic urethroplasty. The anastomosis in two layers ensured that the patients could be discharged from the hospital within 24 h following surgery without a urethral catheter, with a success rate of 90%. This was not inferior to the success rate of 71% when the anastomosis was performed with interrupted sutures. Theoretically, the short catheterization period limits the risk of reduced blood flow and pressure-induced necrosis that may be caused by the delayed presence of a urethral catheter. This study confirmed that early catheter removal has no deleterious effect on the anastomotic urethroplasty outcome. In previous studies, catheterization for seven to twenty-one days was recommended for AU, although three days appear to be adequate. Interestingly, the 24-h post-RGU evaluation exhibited urine extravasation in 67% (or six out of nine) patients [6]. Of note, the urethral catheter used intraoperatively to determine the distal end of the urethral stricture was usually changed to a larger one, usually a 16 F–18 F catheter at the end of the procedure.

The main reason for the routine use of an indwelling urethral catheter is to allow for complete mucosal coaptation in the area of repair. However, a catheter may also simultaneously prevent mucosal cross-healing and urethral distension during voiding. It is possible that if urine extravasates into the surrounding tissue, patients can develop urinomas with fistula formation [6,7]. The extravasation of urine demonstrated in the RUG performed at 24 h post-operatively was limited in the patients of this study, and importantly, they did not develop any complications. A possible explanation for the absence of urinoma formation secondary to the urine extravasation might be the patent urethra. A patent urethra ensures that pressure on the suture line is limited and healing of the anastomotic area is not subject to pressure, except during voiding. Other studies with the two-layer anastomosis technique also reported a low (3%) urine extravasation rate when performing early post-operative voiding cystourethrography (VCUG) [8]. Pericatheter RGU has been recommended to avoid premature catheter removal and subsequent catheterization with unnecessary urethral manipulation [9,10,11]. This study demonstrates that the reinsertion of the Foley catheter may not be required in a patent urethra with limited extravasation.

As stricture length is a contentious issue in anastomotic urethroplasty, only patients who had strictures shorter than 2.5 cm were included. However, Barbagli et al. (2007) could not find a statistical difference in success rates of end-to-end anastomosis in lengths less than 2 cm compared to lengths longer than 2 cm, and Aghaji et al. (2001) reported a success rate of 88 % with a mean length of 3.1 cm [1,11].

Although the etiology was difficult to determine precisely, all of these patients had a purulent urethral discharge. Sexually transmitted diseases (STDs), specifically recurrent gonococcal infection, is still endemic in some parts of the globe, South Africa being one of them. According to our urethral stricture database, 81% of our rural patients have already been treated for STDs (unpublished data). However, none of the patients in our study developed or reported urinary tract sepsis or periurethral abscess. Twenty-three of the 36 (64%) patients presented with a suprapubic cystostomy for the relief of severe obstruction and related septic complications.

The patients who had one DVIU were included in this study. It has to be noted that it has been demonstrated that previously failed DVIU had no negative impact on the outcome of subsequent urethroplasty [12]. Furthermore, multiple failed attempts of dilatation and/or DVIU did not impact the outcome of anastomotic urethroplasty by Eltahawy et al. (2007) [3]. However, for this study, the admission criteria were restricted to one DVIU intervention, as the failure rates of subsequent urethroplasties doubled when patients had a previous manipulation for urethral stricture disease [13].

One patient in Group 2 developed symptoms of stricture recurrence within 1.1 months after urethroplasty due to a poor re-anastomosis technique. If this patient is excluded from the analyses, the success rate of 75% (12/16) does not change our conclusions. The patients in this study were followed up for a minimum of eighteen months, although long-term follow-up is necessary as the success rates of anastomotic urethroplasty can decrease over time, regardless of the technique [14].

Although the sample size was small, the catheterization removal on day one after AU appears to play no role in its outcome in Group 1. The presence of minimal extravasation cleared up without negatively influencing the success rate. Therefore, small amounts of contrast extravasation do not warrant reinsertion of the catheter as long as the patient has a patent urethra.

This study has limitations inherent to local logistics, including the difficult scheduling of and access to uroflowmetry, evaluation of post-void residual urine by ultrasound scan both pre-and post-operatively, as well as quality-of-life issues, including sexual activity. As mentioned above, flexible urethroscopy was mainly reserved for patients with a high suspicion index for stricture recurrence based on obstructive voiding symptoms. Sexual activity remains a rather private issue in rural communities that most patients do not feel comfortable discussing publicly. Therefore, understandably for cultural reasons, this quality-of-life facet was not included in the outcome protocol.

## 5. Conclusions

Anastomotic urethroplasty with a watertight double-layer re-anastomosis and early catheter removal had a cure rate comparable to the conventional interrupted suture anastomosis with 14-day catheterization. The double-layer running suture re-anastomosis allows early discharge and urethral catheter removal and is not associated with an increased restricture rate in the short-to-medium term. This technique warrants further study with larger patient cohorts and a longer follow-up period in patients with bulbar strictures of less than 2.5 cm in length.

## Figures and Tables

**Figure 1 jcm-11-04252-f001:**
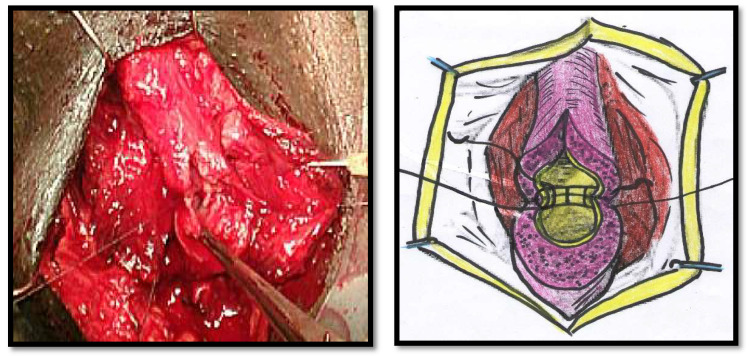
The urethral epithelium sutured with a continuous running suture and the spongiosum sutured in the same fashion separately.

**Figure 2 jcm-11-04252-f002:**
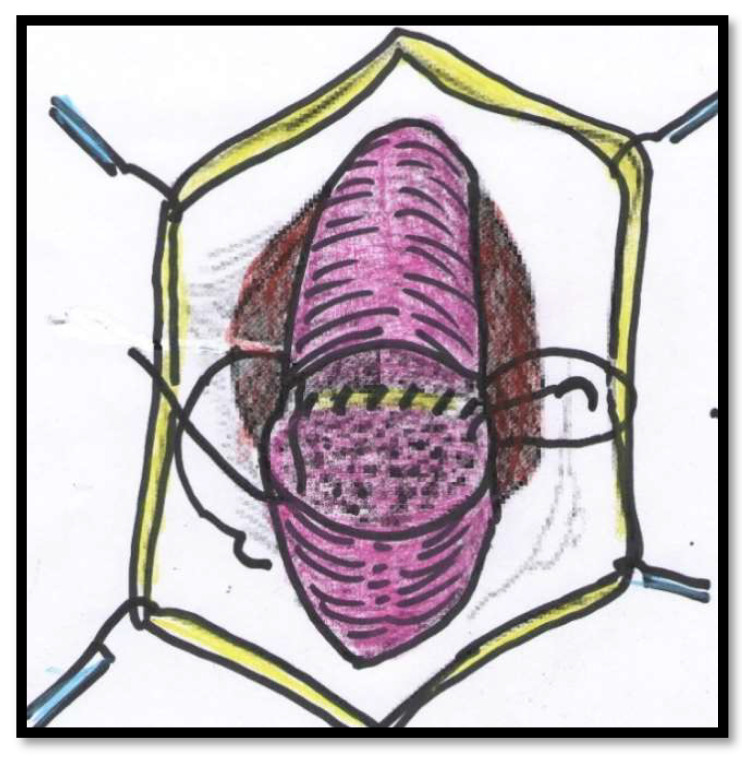
Closing of the spongiosum with a running suture.

**Figure 3 jcm-11-04252-f003:**
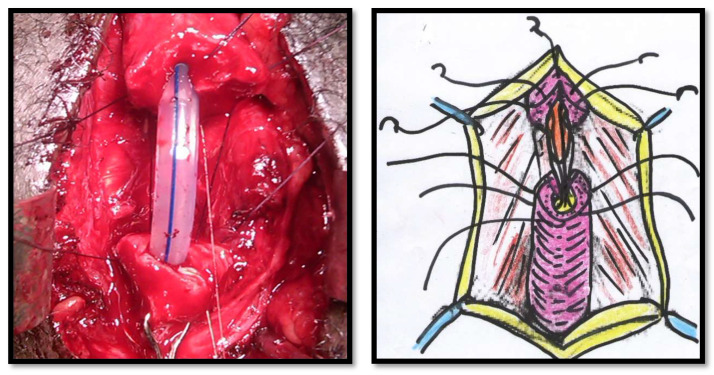
The anastomosis is performed with five interrupted sutures incorporating the epithelium and spongiosum.

**Figure 4 jcm-11-04252-f004:**
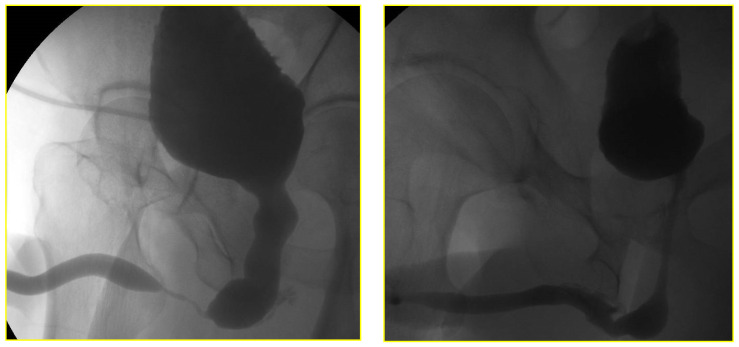
Retrograde urethrogram in a patient with a 2.1 cm stricture pre and 24-h post double-layer running suture re-anastomosis.

**Figure 5 jcm-11-04252-f005:**
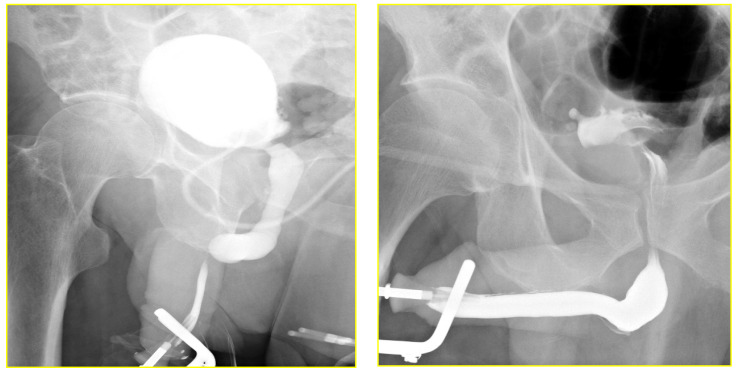
Retrograde urethrogram in a patient with 0.8 cm stricture pre and 24-h post double-layer continuous running suture re-anastomosis.

**Table 1 jcm-11-04252-t001:** Summary of stricture characteristics and surgical outcomes by patient group.

Variable	Group 1	Group 2	*p*-Value
	(*n* = 19)	(*n* = 17)	
Mean age (years)	46.05	45.05	0.822
Mean stricture length (cm)	1.2	1.27	0.631
Success rate (%)	90%	71%	0.219
Time to fail (mean in months)	4.9 (3.6 and 6.2)	5.6 (1.1 to 10.1)	0.804
Mean follow-up (months)	36 (20 to 40)	33 (19 to 41)	0.716

## Data Availability

The lead author (FMC) can be contacted for further details.

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
