# Peer review of "Anastomotic Urethroplasty with Double Layer Continuous Running Suture Re-Anastomosis Versus Interrupted Suture Re-Anastomosis for Infective Bulbar Urethral Strictures: A Prospective Randomised Trial"

_jcm, 2022, doi:10.3390/jcm11154252_

Round 1
Reviewer 1 Report
1. Sentence in abstract A total of thirty-six patients (mean age 45.5 years, range 20 to 69 years) with bulbar urethral strictures were included in this study." is redundant.
2. Vicryl suture . Why do you write vicral?
3. We need a better quality picture instead of Figure 1.1
4. Randomisation and allocation concealment should be better
5. Don't you have uroflow results post procedures?
Author Response
Reviewer 1
- Sentence in abstract A total of thirty-six patients (mean age 45.5 years, range 20 to 69 years) with bulbar urethral strictures were included in this study." is redundant.
Response 1: We agree with the Reviewer and have deleted the sentence.
2. Vicryl suture . Why do you write vicral?
Response 2: We agree with the reviewer. The name is mispelled. It should be Vicryl (polyglactin) absorbable suture. We've extended the correction of this mispelling throughout the manuscript and have highlighted tthe correction in red!
3. We need a better quality picture instead of Figure 1.1
Response 3: A better quality picture will be provided!
4. Randomisation and allocation concealment should be better
Response 4: The patients were randomly allocated to two groups according to their odd and even number of registration in the study: group 1 (G1) included patients with odd numbers and G2 with even numbers according to their arrival at the clinic; that is all odd numbers were allocated to Group 1 and even numbers to Group 2. A prophylactic broad-spectrum antibiotic namely ceftriaxone, was administered preoperatively.
5. Don't you have uroflow results post procedures?
Response 5: I will contact the first author based in South Africa to provide post-op uroflow results as soon as possible.
Reviewer 2 Report
The project research is of interest but I think You can improve this manuscript thus to achieve a better result.
In the abstract is better to report "retrograde urethrography or urethragram" instead of "retrograde urography".
In the abstract, I think is not necessary to repeat either in the "methods" and in the "results" number and type of patients. Please report this data in the results. As example, You could write in the methods: "a consecutive series of patients suffering from urethral bulbar stricture has been enrolled in the study", then in the "Results" you should start with: "36 patients were finally collected for the study". Do not be repetetive.
Minor english revision should be done to provide a more elegant language.
Just to complete the methods, please specify if patients have undergone also flexible urethroscopy in the preoperative diagnostic algorythm or in the postoperative one to check the surgical outcome.
Probably You could add some data regarding patient's cohort:
- etiology of urethral stricture, time from urethral trauma or time from diagnosis, time from first surgical approach (for patients already submitted to endoscopic surgery),
- clinical significant events (urosepsis) before surgery
- patients with cystostomy (number).
- instrumental data (if you have): uroflowmetry, pre and postoperative, post residual urine volume, symptomatic scores, including QoL (if you have), data regarding sexual activity (how many patients were sexually active before surgery ? after treatment has this number improved ?).
Functional data are important because not only the significant clinical events (urinary retention or infections) indicate post operative troubles. If in the postoperative assessment you're going to observe urinary flow reduction this can be an early sign of complication, especially if some additional data (ultrasound) show that lower urinary tract is suffering from bladder outflow obstruction. In this protocol you report only a six weeks postoperative RGU as evaluation tool.
Please specify in the "methods" if the final catheter was the same that intraoperative, indicating type of catheter (material) and Fr.
Did You approach surgery using optical devices (magnification glasses / microscope ?). Was the firs surgeon always the same ?
Which instruments did you use for coagulation (monopolar, bipolar) ?
Did you administrate steroids (pre, intra, or postoperative ?)
Try to enrich of clinical and methodological data the manuscript to better support the results and the discussion.
Author Response
Thank you for your comments and corrections.
Our responses are as follows:
1. In the abstract is better to report "retrograde urethrography or urethragram" instead of "retrograde urography". Yes, we have corrected this terminology as you suggest. The term "retrograde urethrography or urethragram" has been replaced the term urography".
2. In the abstract, I think is not necessary to repeat either in the "methods" and in the "results" number and type of patients. Please report this data in the results. As example, You could write in the methods: "a consecutive series of patients suffering from urethral bulbar stricture has been enrolled in the study", then in the "Results" you should start with: "36 patients were finally collected for the study". Do not be repetetive. Your suggested corrections have been included in the abstract. I would like to mention that we had already addressed this matter as Reviewer 1 had also suggested the same correction which we did. We have already submitted separately the revisions requested by Reviewer 1 yesterday. Unfortunately, for urgent matters (emergency on-call work) we had to interrupt and postpone for today the responses to Reviewer 2.
3. Minor english revision should be done to provide a more elegant language. We will definitely do the English language corrections with a native English language teacher before submission of the final version after completing all the technical corrections.
4. Just to complete the methods, please specify if patients have undergone also flexible urethroscopy in the preoperative diagnostic algorythm or in the postoperative one to check the surgical outcome. Yes, all patients underwent 16F flexible urethroscopy for routinely pre-operative and post-operative assessment. We have included that in the the abstract and manuscript.
5. Probably You could add some data regarding patient's cohort:
- etiology of urethral stricture, time from urethral trauma or time from diagnosis, time from first surgical approach (for patients already submitted to endoscopic surgery),
- clinical significant events (urosepsis) before surgery
- patients with cystostomy (number).
instrumental data (if you have): uroflowmetry, pre and postoperative, post residual urine volume, symptomatic scores, including QoL (if you have), data regarding sexual activity (how many patients were sexually active before surgery ? after treatment has this number improved ?). We will discuss and obtain all this information with the 1st author who is based in Bloemfontein. I am a co-author, Francisco Martins, who has advised and assisted the other co-authors with the surgical procedures. All this information will be collected for the subsequent, eventually final, version of the article before approval.
6. Functional data are important because not only the significant clinical events (urinary retention or infections) indicate post operative troubles. If in the postoperative assessment you're going to observe urinary flow reduction this can be an early sign of complication, especially if some additional data (ultrasound) show that lower urinary tract is suffering from bladder outflow obstruction. In this protocol you report only a six weeks postoperative RGU as evaluation tool. Yes, we agree completely! As said above, we will provide these data soon.
7. Please specify in the "methods" if the final catheter was the same that intraoperative, indicating type of catheter (material) and Fr. No, the intraoperative catheter was in the majority of patients a 6-8F urethral or ureteric catheter placed over a guidewire not to lose urethral continuity. In total obliterative strictures, a 14-16F Foley was used to identify the distal end of the strictured area. Postoperatively, a 18F Sylastic catheter was used for about 3 weeks. Modification included in the abstract text.
8. Did You approach surgery using optical devices (magnification glasses / microscope ?). Was the firs surgeon always the same ? No. The authors occasionally use magnifying lens when flaps need to be harvested to avoid damage to flap blood supply. Yes, the first surgeon was always the same (Freddie Claasen).
9. Which instruments did you use for coagulation (monopolar, bipolar)? We use bipolar and monopolar indifferently in these cases, mostly monopolar. We tend to use bipolar for harvesting flaps with delicate pedicles and blood supply.
10. Did you administrate steroids (pre, intra, or postoperative ?) No, we don't use steroids postoperatively in these patients.
11. Try to enrich of clinical and methodological data the manuscript to better support the results and the discussion. Yes, we absolutely with the Reviewer's suggestion/recommendation. We will definitely do so int he next version of the manuscript.
We have uploaded the file with some of the corrections already included as per the Reviewer 2's request. We will correct the whole manuscript soon and will resubmit to JCM Editorial Team!
Thank you for your kind and helpful support and suggestions.
Best regards,
Francisco Martins
